

# A retrospective analysis of neoadjuvant chemotherapy in pneumonectomy for locally advanced central non–small cell lung cancer

Yuchen Wang*, Zhihang Dang*, Pu Jiang, Zhifeng Li, Jin Yang, Kun Gao, Xiaona Chen and Jifang Yao

Department of Thoracic Surgery, The Fourth Hospital of Hebei Medical University, Shijiazhuang, China
* These authors contributed equally to this work.

Corresponding author
Jifang Yao, yaojifang2023@126.com

## ABSTRACT

**Aims:** The treatment of patients with locally advanced central non–small cell lung cancer (NSCLC) remains controversial. This study aimed to evaluate the effects of neoadjuvant chemotherapy in patients undergoing pneumonectomy.

**Methods:** A retrospective analysis was conducted on patients who underwent pneumonectomy with or without neoadjuvant chemotherapy for locally advanced central NSCLC between 2014 and 2019. Categorical variables were compared using the Chi-square or Fisher's exact test. Survival analysis was performed using the Kaplan–Meier method, with comparisons made *via* the log-rank test. Multivariate analysis of independent prognostic factors was conducted using the Cox proportional hazards regression model. A $p$-value < 0.05 was considered statistically significant.

**Results:** Based on inclusion and exclusion criteria, 104 patients were selected from a total of 6,930, including 69 who received neoadjuvant chemotherapy and 35 who did not. Univariate analysis showed that the neoadjuvant chemotherapy group had significantly improved 5-year overall survival (OS: 29.1% *vs.* 12.8%, $\chi^2$ = 4.089, $p$ = 0.043) and disease-free survival (DFS: 22.3% *vs.* 8.8%, $\chi^2$ = 3.941, $p$ = 0.047). The downstaging rate in the neoadjuvant chemotherapy group was 29.0%. Subgroup analysis revealed that patients with downstaging had significantly better 5-year OS and DFS compared to those without downstaging (OS: 56.6% *vs.* 17.1%, $\chi^2$ = 10.266, $p$ = 0.001; DFS: 54.1% *vs.* 6.0%, $\chi^2$ = 20.785, $p$ < 0.001). Another subgroup analysis showed that although 5-year DFS was 0% in both groups, patients with stage cN2 disease who received neoadjuvant chemotherapy had better 5-year OS (16.3% *vs.* 7.8%, $\chi^2$ = 5.603, $p$ = 0.018) and a statistically significant difference in DFS ($\chi^2$ = 7.328, $p$ = 0.007).

**Conclusions:** Neoadjuvant chemotherapy significantly improves prognosis in patients with locally advanced central NSCLC undergoing pneumonectomy. Multivariate analysis confirms its positive impact on survival. Patients who experience downstaging after neoadjuvant chemotherapy show notably better outcomes. For patients with stage cN2 disease, neoadjuvant chemotherapy is associated with improved survival.

## INTRODUCTION

Locally advanced lung cancer accounts for about 30% of lung cancer. In order to achieve long-term survival, although some patients can undergo sleeve lobectomy, sometimes pneumonectomy is inevitable for patients with central lung cancer (*Graham & Singer, 1934*; *Ferguson & Lehman, 2003*). Furthermore, evidence indicates that preoperative induction therapy, such as neoadjuvant chemotherapy and neoadjuvant radiotherapy, is recommended to improve the prognosis of patients with locally advanced non-small cell lung cancer (NSCLC) (*Martins et al., 2012*; *Song et al., 2010*). For patients with locally advanced central NSCLC, due to the complications and perioperative mortality of pneumonectomy and the side effects of induction therapy (*Brunelli et al., 2020*; *Albain et al., 2009*), even though recent studies reported that patients can benefit from induction therapy and pneumonectomy after induction therapy is safe (*Sonett et al., 2004*; *Weder et al., 2010*; *Arame, Mordant & Riquet, 2014*; *Mansour et al., 2007*; *Evans et al., 2010*; *Gaissert et al., 2009*), many centers avoid induction therapy before pneumonectomy. The treatment of patients with locally advanced central NSCLC is still controversial. Therefore, we conducted this retrospective analysis to evaluate the long-term outcomes of patients with neoadjuvant chemotherapy before pneumonectomy.

## MATERIALS AND METHODS

### Patients

A retrospective analysis was performed of all patients who underwent radical pneumonectomy with or without neoadjuvant chemotherapy for locally advanced central non–small cell lung cancer at the Fourth Hospital of Hebei Medical University from 2014 to 2019. This study was approved by the ethics committee of The Fourth Hospital of Hebei Medical University (2022MECD58). The informed consent before surgery included permission to perform such an outcome analysis. The clinical and pathological data of all patients were complete.

Patient enrollment criteria: (1) Type as central lung cancer; (2) Radical pneumonectomy; (3) Postoperative pathological type of non-small cell lung cancer; (4) The clinical stage of patients in neoadjuvant chemotherapy group was II-III, neoadjuvant chemotherapy (pemetrexed + platinum, paclitaxel + platinum) for 2–3 cycles. The pathological stage of patients in the surgery group was II-III. (5) Cardiac, pulmonary, hepatic, renal functions and complete blood examination were normal and could tolerate neoadjuvant chemotherapy and surgery. Exclusion criteria: (1) Resection of R1 and R2; (2) Perioperative mortality; (3) Patients who could not tolerate neoadjuvant chemotherapy; (4) Patients with clinical or pathological staged N3. (5) patients in the neoadjuvant chemotherapy group who had received other neoadjuvant therapy before surgery. (6) Patients whose medical records show that no subsequent surgical treatment was performed after neoadjuvant chemotherapy therapy.

## Diagnosis and treatment

Tumors were classified and staged according to the eighth edition of tumor node metastasis (TNM) staging for lung cancer (*Detterbeck et al., 2017*). Clinical staging and restaging were performed with positron emission tomography integrated (PET) in computed tomography (CT) or CT of the head, chest and upper abdomen. Additional staging tests, such as brain magnetic resonance imaging and bone scintigraphy, were used according to clinical signs and symptoms. Bronchoscopic biopsy or endobronchial ultrasound guided transbronchial needle aspiration was performed to obtain a preoperative pathological type for neoadjuvant chemotherapy. The pathological type was confirmed by histology or cytology.

The neoadjuvant chemotherapy group received pemetrexed (500 mg/m$^2$, intravenous drip, day 1) and cisplatin (75 mg/m$^2$, intravenous drip, day 1) every 3–4 weeks or paclitaxel (175 mg/m$^2$, intravenous drip, day 1) and cisplatin (75 mg/m$^2$, intravenous drip, day 1) every 3–4 weeks. Neoadjuvant chemotherapy was performed for 2–3 cycles before surgery, and surgery was performed 4–6 weeks after chemotherapy. Tumors responses were evaluated using the Response Evaluation Criteria in Solid Tumor (RECIST) guidelines (*Eisenhauer et al., 2009*). Patients underwent thoracotomy or video-assisted thoracoscopy with resection pneumonectomy and systematic mediastinal lymph node dissection.

## Follow-up

Complete follow-up information including imaging, operative, and pathologic reports, was achieved for each patient concerning survival and recurrence. Follow-up visits were scheduled every 3 months for 1 year, every 6 months for the next 4 years, and then annually.

## Statistical analysis

Statistical analysis was conducted using the SPSS statistical software (version 23.0; IBM Corp., Armonk, NY, USA). Frequencies were compared with the Chi-square or Fisher's exact test for categorical variables. Overall survival (OS) was calculated from the first day of surgery until death from any cause or the date of last follow-up. Disease-free survival (DFS) was calculated from the date of surgery to the date of recurrence or metastasis. Survival analysis was performed using the Kaplan–Meier method and curves were compared with the log–rank test. The Cox proportional hazards regression model performed multivariate analysis of independent prognostic factors. $p < 0.05$ was considered as significant.

## RESULTS

### Study population description

According to the enrollment and exclusion criteria, 104 patients were enrolled from 6,930 patients in this study, including 69 patients receiving neoadjuvant chemotherapy before pneumonectomy and 35 patients undergoing pneumonectomy without neoadjuvant chemotherapy. Baseline clinical characteristics can be found in Table 1.

**Table 1  Baseline clinical characteristics of the study population.**

| Characteristics | Neoadjuvant chemotherapy (n = 69) | Pneumonectomy (n = 35) | $\chi^2$ | p |
|---|---|---|---|---|
| Gender (n%) | | | 1.351 | 0.25 |
| Male | 60 (87.0) | 33 (94.3) | | |
| Female | 9 (13.0) | 2 (5.7) | | |
| Age (n%) | | | 0.134 | 0.71 |
| ≤60 years | 42 (60.9) | 20 (57.1) | | |
| >60 years | 27 (39.1) | 15 (42.9) | | |
| Pathological type (n%) | | | 0.066 | 0.8 |
| SCC | 49 (71.0) | 24 (68.6) | | |
| ADC | 20 (29.0) | 11 (31.4) | | |
| Laterality (n%) | | | 1.492 | 0.22 |
| Left | 52 (75.4) | 30 (85.7) | | |
| Right | 17 (24.6) | 5 (14.3) | | |
| cT (n%) | | | 2.898 | 0.24 |
| 2 | 24 (34.8) | 18 (51.4) | | |
| 3 | 27 (39.1) | 9 (25.7) | | |
| 4 | 18 (26.1) | 8 (22.9) | | |
| cN (n%) | | | 0.055 | 0.97 |
| 0 | 26 (37.7) | 14 (40.0) | | |
| 1 | 10 (14.5) | 5 (14.3) | | |
| 2 | 33 (47.8) | 16 (45.7) | | |
| cTNM (n%) | | | 0.450 | 0.5 |
| 2 | 23 (33.3) | 14 (40.0) | | |
| 3 | 46 (66.6) | 21 (60.0) | | |

Note:
ADC, adenocarcinoma; SCC, squamous cell carcinoma; cN, clinical N-stage; cT, clinical T-stage; cTNM, clinical TNM-stage. Categorical variables were presented as frequency (%). Chi-square test was used to compare the general information between the patients with neoadjuvant chemotherapy and pneumonectomy.

## Evaluation of patients with locally advanced central lung cancer after neoadjuvant chemotherapy and pneumonectomy

The postoperative pathologic features are shown in Table 2. According to the postoperative pathology reports, six patients had vascular invasion. As a consequence of neoadjuvant chemotherapy, 20 (29.0%) patients experiences downstaging. However, no pathologic complete responses were observed.

## Overall survival and disease-free survival

In order to further explore the effect of neoadjuvant chemotherapy on long-term outcomes, we first examined the 5-year OS and 5-year DFS of all patients, which were 24.9% and 18.5%, respectively (Fig. 1). The univariate analysis showed all variables impact on survival in Table 3.

After that, we performed an in-depth analysis of the 5-year OS and 5-year DFS in different groups by Kaplan-Meier analysis. As shown in Fig. 2A, the 5-year OS was 29.1% in the neoadjuvant chemotherapy group and 12.8% in the pneumonectomy alone group

**Table 2 The prognostic pathological features of neoadjuvant chemotherapy group and pneumonectomy group.**

| Characteristics | Neoadjuvant chemotherapy ($n$ = 69) | Pneumonectomy ($n$ = 35) |
|---|:---:|:---:|
| ypT(pT) ($n$%) | | |
| 1 | 32 (46.4) | 0 (0.0) |
| 2 | 17 (24.6) | 18 (51.4) |
| 3 | 4 (5.8) | 9 (25.7) |
| 4 | 16 (23.2) | 8 (22.9) |
| ypN(pN) ($n$%) | | |
| 0 | 28 (40.6) | 14 (40.0) |
| 1 | 8 (11.6) | 5 (14.3) |
| 2 | 33 (47.8) | 16 (45.7) |
| ypTNM(pTNM) ($n$%) | | |
| 1 | 16 (23.2) | 0 (0.0) |
| 2 | 11 (15.9) | 14 (40.0) |
| 3 | 42 (60.9) | 21 (60.0) |
| Vessel invasion ($n$%) | | |
| Yes | 2 (2.9) | 4 (11.4) |
| No | 67 (97.1) | 31 (88.6) |
| Downstaging ($n$%) | | |
| Yes | 20 (29.0) | – |
| No | 49 (31.0) | – |

Note:
ypT, post-neoadjuvant pathologic T-stage; ypN, post-neoadjuvant pathologic N-stage; ypTNM, post-neoadjuvant pathologic TNM-stage; pT, pathologic T-stage; pN, pathologic N-stage; pTNM, pathologic TNM-stage.

($\chi^2$ = 4.089, $p$ = 0.043). At the same time, we found the rate of 5-year DFS was significantly higher in the neoadjuvant chemotherapy group (22.3% $vs.$ 8.8%, $\chi^2$ = 3.941, $p$ = 0.047, Fig. 2B).

Further analysis in the subgroup of 69 patients with neoadjuvant chemotherapy, showed that the 5-year OS was 56.6% in the group of patients with downstaging and 17.1% in the group of patients with non-downstaging ($\chi^2$ = 10.266, $p$ = 0.001, Fig. 3A). Similar results are also reflected in DFS, where the results showed the 5-year DFS was 54.1% in the group of patients with downstaging and 6.0% in the group of patients with non-downstaging ($\chi^2$ = 20.785, $p$ = 0.000, Fig. 3B).

Just like the results above all, the results in the subgroup of 49 patients with stage cN2 showed that the 5-year OS was 16.3% in the group with neoadjuvant chemotherapy and 7.8% in the group with pneumonectomy alone ($\chi^2$ = 5.603, $p$ = 0.018, Fig. 4A). Meanwhile, it is worth noting that although the 5-year DFS in both groups was 0%, the difference between groups was statistically significant ($\chi^2$ = 7.328, $p$ = 0.007, Fig. 4B).

In order to verify the effect of neoadjuvant chemotherapy, we conducted a multivariate analysis. The independent effect on survival of significant variables was analyzed using the forward stepwise cox regression. Since the response to neoadjuvant chemotherapy, downstaging, and post-neoadjuvant TNM stage were highly correlated, the preoperative

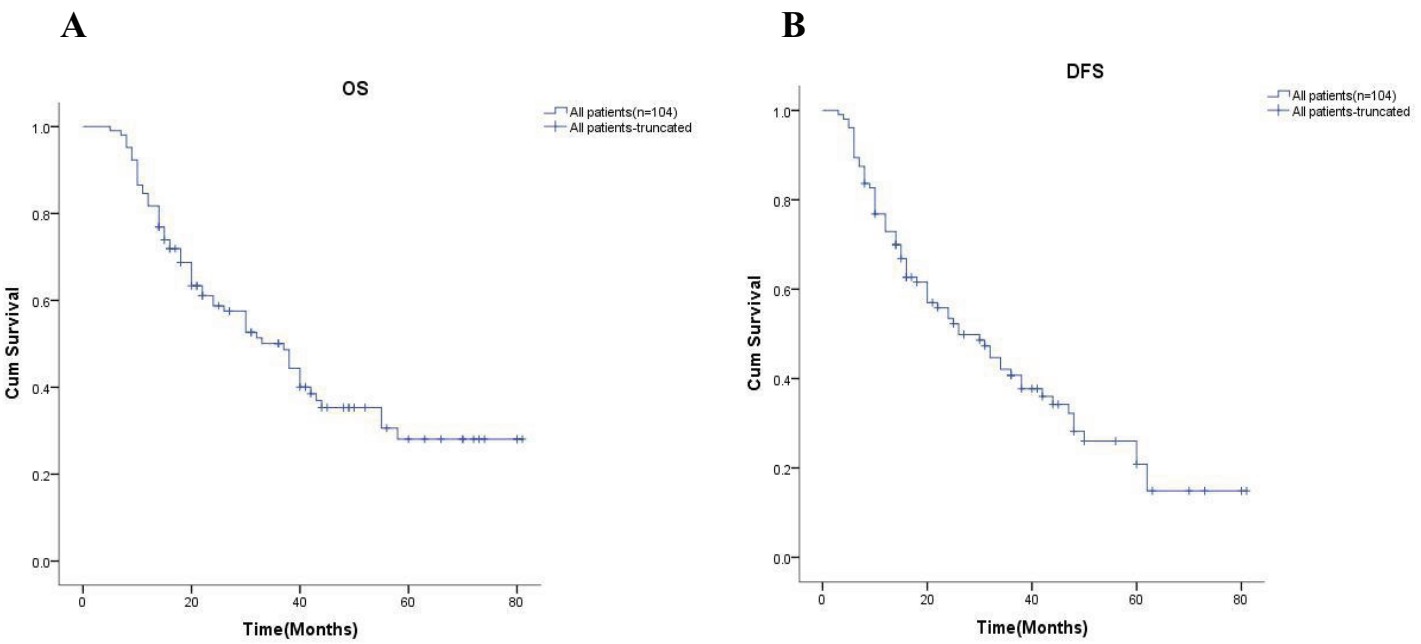

**Figure 1 Kaplan-Meier analysis of all patients.** (A) and (B) showed the 5-year OS rate and the 5-year DFS rate for all patients respectively. OS, overall survival; DFS, disease-free survival.

and postoperative variables were tested separately. The results of the multivariate analysis are presented in Tables 4 and 5. In preoperative variables, our results showed statistically significant differences in OS and DFS between patients who received neoadjuvant chemotherapy and those who did not. In particular, OS and DFS in patients with stage cT4 or stage cN2 were significantly worse than in others. In postoperative variables, both OS and DFS in patients with stage ypT (pT) 3–4 or stage ypN (pN) 2 were significantly worse than in others. As for vessel invasion, it only had an impact on the OS of patients.

## DISCUSSION

The treatment of patients with locally advanced NSCLC remains a challenge and is discussed controversially especially for central lung cancer. Generally, sleeve lobectomy can bring better long-term survival and quality of life with the progress of surgery (*Magouliotis et al., 2022*), in some cases such as invasion of the tumor into the great vessels or main stem bronchi, we have to undergo pneumonectomy for a radical resection (*Graham & Singer, 1934*; *Ferguson & Lehman, 2003*; *Weder et al., 2010*; *Arame, Mordant & Riquet, 2014*). Recent studies have shown that pneumonectomy is safe after induction therapy and preoperative induction therapy can downstage tumors or improve the prognosis of patients (*Martins et al., 2012*; *Song et al., 2010*; *Sonett et al., 2004*; *Gaissert et al., 2009*). Thus, we have great expectations for preoperative induction therapy, like neoadjuvant chemotherapy and neoadjuvant radiotherapy, before surgery. Based on the results of two studies, compared with neoadjuvant chemotherapy, neoadjuvant chemoradiotherapy did not improve the survival of patients (*Thomas et al., 2008*;

**Table 3 The univariate analysis of variables for 5-year overall survival and 5-year disease-free survival.**

| Variables | n | Overall survival | | | Disease-free survival | | |
|---|---|---|---|---|---|---|---|
| | | 5-year survival (%) | $\chi^2$ | p | 5-year disease-free survival (%) | $\chi^2$ | p |
| Gender | | | 2.682 | 0.102 | | 0.160 | 0.690 |
| Male | 93 | 21.0 | | | 19.5 | | |
| Female | 11 | 56.1 | | | 13.6 | | |
| Age | | | 0.147 | 0.701 | | 0.343 | 0.558 |
| ≤60 years | 62 | 26.1 | | | 17.0 | | |
| >60 years | 42 | 22.8 | | | 23.3 | | |
| Pathological type | | | 1.525 | 0.217 | | 3.831 | 0.050 |
| SCC | 73 | 25.2 | | | 25.5 | | |
| ADC | 31 | 22.2 | | | 0.0 | | |
| Laterality | | | 0.309 | 0.578 | | 0.221 | 0.638 |
| Left | 82 | 25.7 | | | 21.4 | | |
| Right | 22 | 21.9 | | | 9.5 | | |
| cT | | | 24.129 | 0.000 | | 14.249 | 0.000 |
| 2 | 42 | 31.4 | | | 12.2 | | |
| 3 | 36 | 32.1 | | | 33.3 | | |
| 4 | 26 | 6.6 | | | 11.0 | | |
| cN | | | 11.436 | 0.001 | | 22.386 | 0.000 |
| 0 | 40 | 40.9 | | | 37.4 | | |
| 1 | 15 | 28.7 | | | 31.5 | | |
| 2 | 49 | 12.5 | | | 0.0 | | |
| cTNM | | | 19.350 | 0.000 | | 32.464 | 0.000 |
| 2 | 37 | 47.5 | | | 46.3 | | |
| 3 | 67 | 12.4 | | | 3.4 | | |
| ypT(pT) | | | 49.974 | 0.000 | | 39.269 | 0.000 |
| 1 | 32 | 43.7 | | | 31.0 | | |
| 2 | 35 | 26.7 | | | 19.1 | | |
| 3 | 13 | 0.0 | | | 20.5 | | |
| 4 | 24 | 0.75 | | | 12.1 | | |
| ypN(pN) | | | 10.656 | 0.001 | | 21.093 | 0.000 |
| 0 | 42 | 37.4 | | | 35.1 | | |
| 1 | 13 | 33.6 | | | 37.6 | | |
| 2 | 49 | 12.5 | | | 0.0 | | |
| ypTNM(pTNM) | | | 21.793 | 0.000 | | 34.486 | 0.000 |
| 1 | 16 | 62.5 | | | 57.7 | | |
| 2 | 25 | 35.7 | | | 35.6 | | |
| 3 | 63 | 10.6 | | | 0.0 | | |
| Group | | | 4.089 | 0.043 | | 3.941 | 0.047 |
| Neoadjuvant chemotherapy | 69 | 29.1 | | | 22.3 | | |
| Pneumonectomy | 35 | 12.8 | | | 8.8 | | |
| Vessel invasion | | | 8.208 | 0.004 | | 0.804 | 0.370 |
| Yes | 6 | 0.0 | | | 33.3 | | |
| No | 98 | 26.2 | | | 18.8 | | |

**Note:**
ADC, adenocarcinoma; SCC, squamous cell carcinoma; cN, clinical N-stage; cT, clinical T-stage; cTNM, clinical TNMstage; ypT, post-neoadjuvant pathologic T-stage; ypN, post-neoadjuvant pathologic N-stage; ypTNM, post-neoadjuvant pathologic TNM-stage.

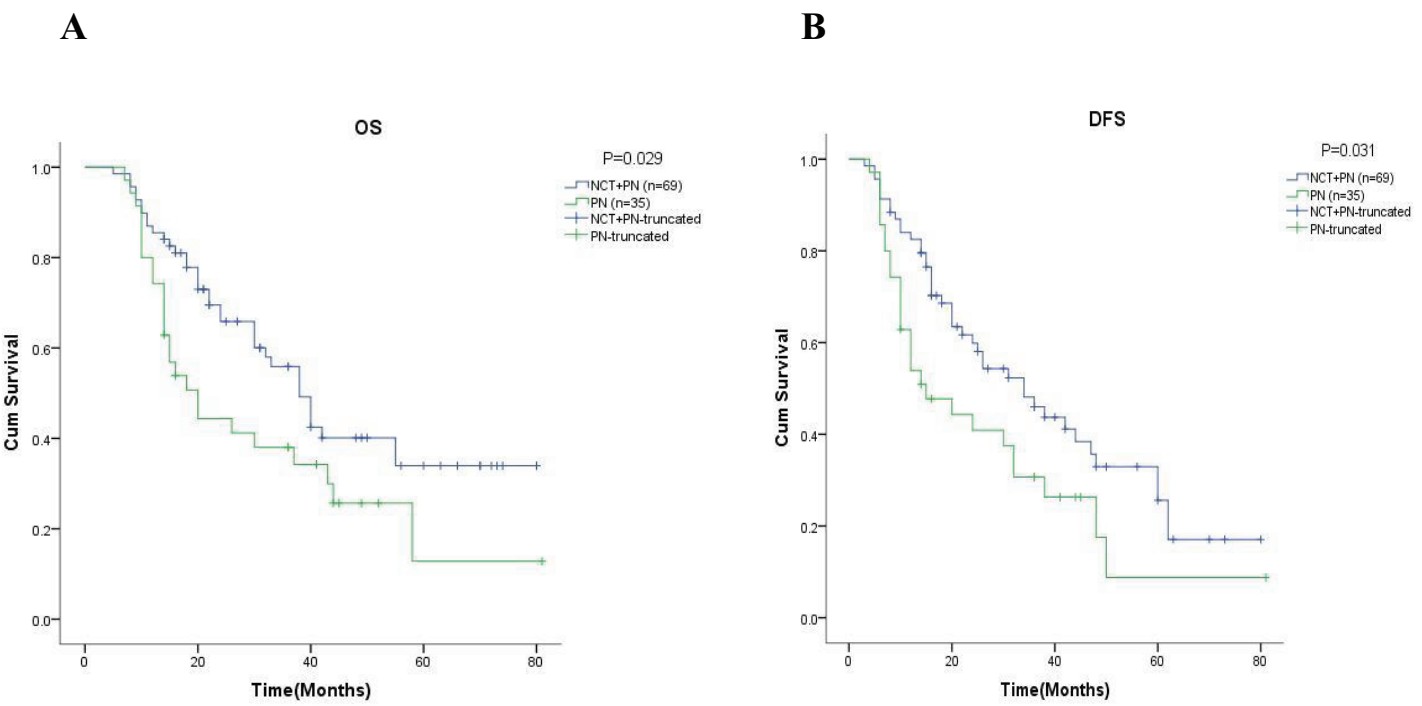

**Figure 2 Kaplan-Meier analysis of patients in different groups.** (A) and (B) were showed the 5-year OS rate and the 5-year DFS rate respectively. NCT, neoadjuvant chemotherapy; PN, pneumonectomy.

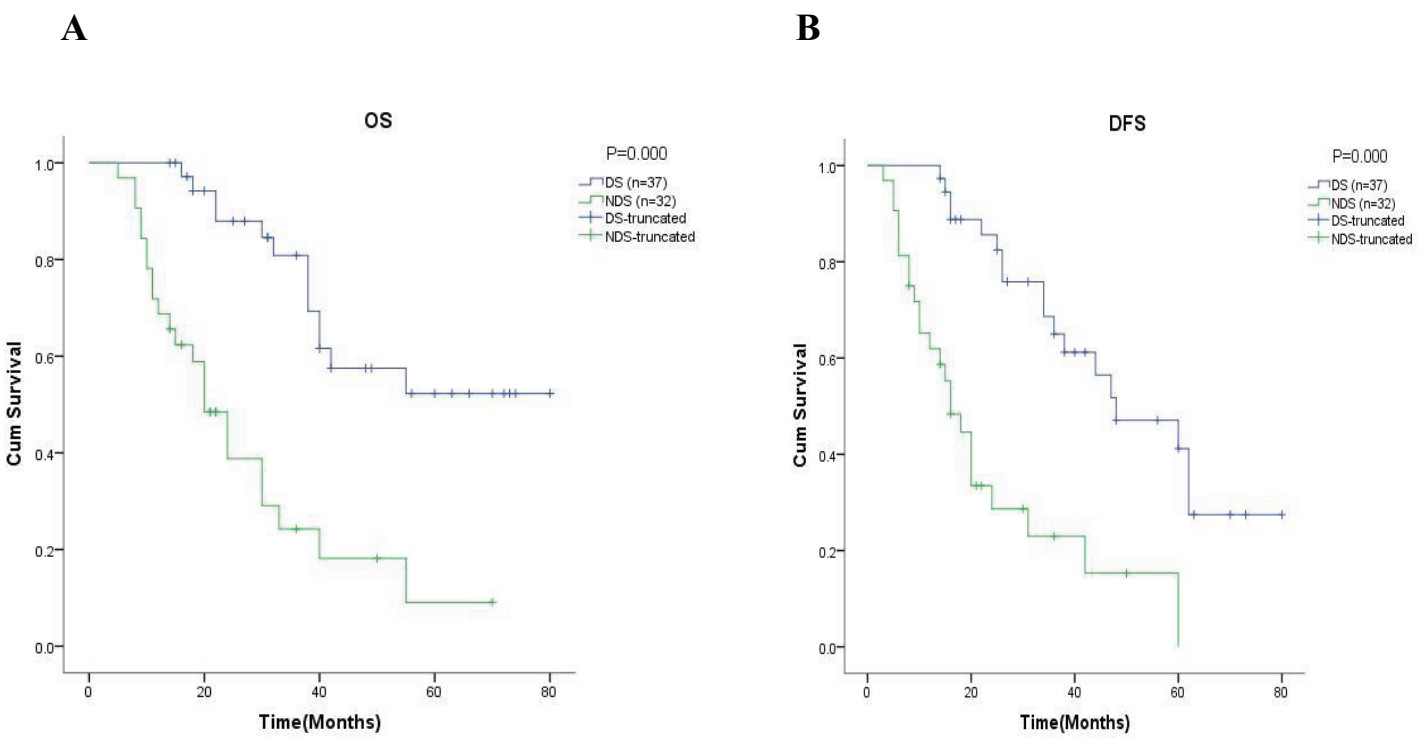

**Figure 3 Kaplan-Meier analysis of patients in neoadjuvant chemotherapy group.** The 5-year OS and the 5-year DFS rate of patients in the neoadjuvant chemotherapy group were showed in (A) and (B) respectively. DS, downstaging; NDS, non-downstaging.

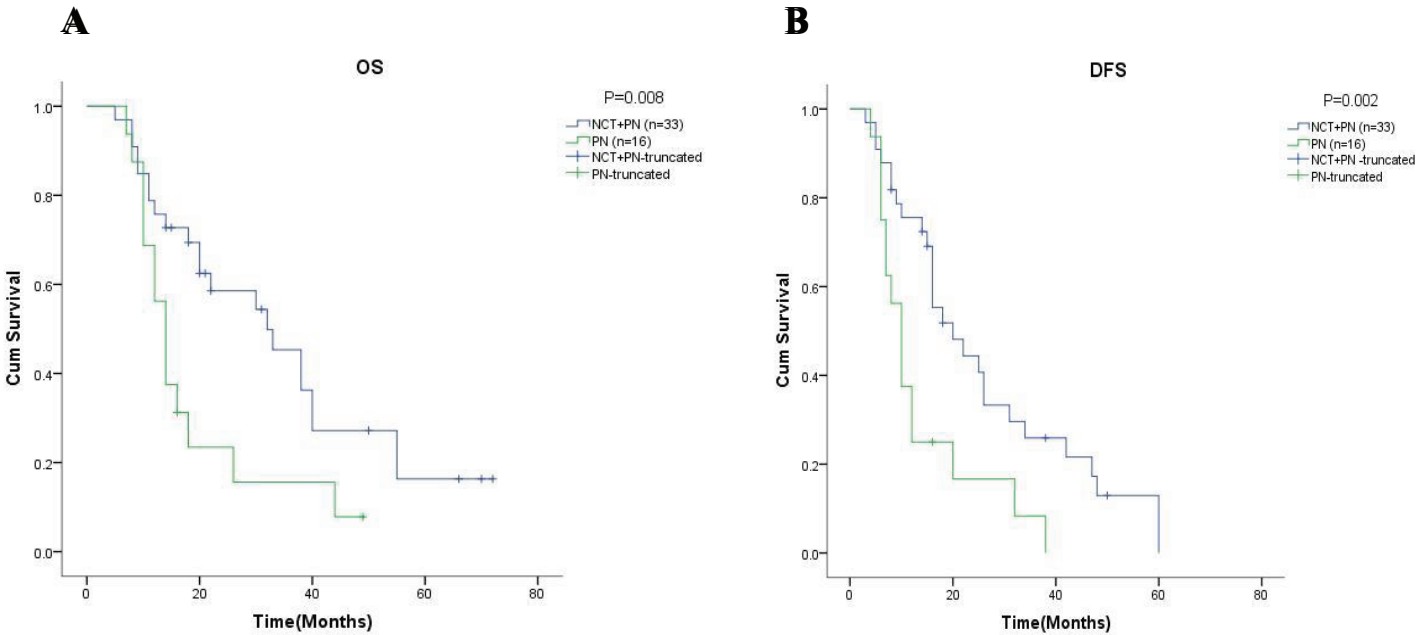

**Figure 4 Kaplan-Meier analysis of patients with clinical N2-stage (cN2).** (A) and (B) described the rate of 5-year OS and 5-year DFS of patients with clinical N2 stage respectively. NCT, neoadjuvant chemotherapy; PN, pneumonectomy.               

**Table 4 Multivariate analysis of preoperative variables.**

| Variables | Overall survival | | | Disease-free survival | | |
|---|---|---|---|---|---|---|
| | HR | 95.0% CI | *p* | HR | 95.0% CI | *p* |
| Group | | | | | | |
| Pneumonectomy | Ref. | | | Ref. | | |
| Neoadjuvant chemotherapy | 0.447 | [0.261–0.768] | 0.003 | 0.419 | [0.246–0.713] | 0.001 |
| cT | | | | | | |
| 2 | Ref. | | | Ref. | | |
| 3 | 1.498 | [0.777–2.890] | 0.228 | 1.282 | [0.688–2.389] | 0.433 |
| 4 | 10.070 | [4.939–20.532] | 0.000 | 8.526 | [4.299–16.906] | 0.000 |
| cN | | | | | | |
| 0 | Ref. | | | Ref. | | |
| 1 | 1.965 | [0.894–4.319] | 0.093 | 2.281 | [1.036–5.023] | 0.041 |
| 2 | 3.273 | [1.798–5.958] | 0.000 | 4.830 | [2.639–8.841] | 0.000 |

**Note:**
   HR, Hazard Ratio; CI, confidence interval; Ref., Reference; cT, clinical T-stage; cN, clinical N-stage.

*Shah et al., 2012*), our retrospective study was to demonstrate that patients with neoadjuvant chemotherapy before pneumonectomy can achieve better survival.

In a randomized phase III-intergroup trial, major response to chemotherapy was reported in 41% of patients (*Pisters et al., 2010*), According to previous studies, neoadjuvant radiotherapy combined with chemotherapy can improve pathological outcomes compared with neoadjuvant chemotherapy in patients with stage III lung cancer,

**Table 5 Multivariate analysis of postoperative variables.**

| Variables | Overall survival | | | Disease-free survival | | |
|---|---|---|---|---|---|---|
| | HR | 95.0% CI | *p* | HR | 95.0% CI | *p* |
| ypT(pT) | | | | | | |
| 1 | Ref. | | | Ref. | | |
| 2 | 1.745 | [0.852–3.573] | 0.128 | 1.457 | [0.783–2.713] | 0.235 |
| 3 | 6.625 | [2.565–17.112] | 0.000 | 5.947 | [2.452–14.423] | 0.000 |
| 4 | 18.708 | [8.007–43.708] | 0.000 | 14.659 | [6.716–31.994] | 0.000 |
| ypN(pN) | | | | | | |
| 0 | Ref. | | | Ref. | | |
| 1 | 1.348 | [0.587–3.095] | 0.481 | 1.663 | [0.727–3.805] | 0.228 |
| 2 | 3.996 | [2.149–7.431] | 0.000 | 5.591 | [3.043–10.270] | 0.000 |
| Vessel invasion | | | | | | |
| No | Ref. | | | – | – | – |
| Yes | 4.182 | [1.548–11.229] | 0.005 | – | – | – |

Note:
HR, Hazard Ratio; CI, confidence interval; Ref., Reference; ypT, post-neoadjuvant pathologic T-stage; ypN, post-neoadjuvant pathologic N-stage.

but there is no significant improvement in overall survival (*Sher et al., 2015*). These studies have shown that neoadjuvant chemotherapy is effective. In our study, downstaging occurred in 20 (29.0%) patients in neoadjuvant chemotherapy group with partial responses or major pathologic responses. Since our study excluded patients who could not tolerate the operation after chemotherapy and some of the patients with progressive disease, these may be the reasons for the different proportion of downstaging. Anyway, based on our results, we have reason to believe that neoadjuvant chemotherapy is beneficial in these patients.

For the response of induction therapy, the rate of pathological complete response up to 21% in patients with neoadjuvant chemoradiotherapy reported by *Weder et al. (2010)*. Likewise, other research has shown that neoadjuvant chemoradiotherapy contribute to increases pathological response and mediastinal downstaging compared with neoadjuvant chemotherapy (*Thomas et al., 2008*). In our study, complete responses or complete pathological responses were not observed, which was inconsistent with previous reports, maybe owing to the patients received preoperative neoadjuvant chemotherapy alone.

The univariate analysis of the subgroups, we found that patients with downstaging compared to non-downstaging was related to higher 5-year survival and 5-year disease-free survival. Although there is no significance in multivariate analysis, whether these patients who benefit from chemotherapy can achieve long-term survival is still worthy of further study.

Previous study has shown that neoadjuvant chemotherapy is effective for the patients with N2 non-small cell lung cancer (*Stefani et al., 2010*). Consistent results were obtained in our study, even though multivariate COX regression analysis of preoperative variables showed that the OS and DFS were significantly worse in patients with stage cN2, compared

with the pneumonectomy group, neoadjuvant chemotherapy improved OS and DFS. The research about neoadjuvant chemotherapy by *Decaluwé et al. (2009)* demonstrated that patients with mediastinal nodal downstaging compared to patients with persistent N2 disease received a better survival, and for the classification of ypN category, compared to multilevel-ypN2 and ypN3, ypN0-1 and ypN2-single level have a better survival in multivariate analysis. Interestingly, by multivariate COX analysis of postoperative variables, our analysis corroborated similar findings that OS and DFS were significantly worse in patients with stage ypN (pN) 2. For patients with persistent N2 disease after induction therapy, although the prognosis is worse, surgical treatment is still necessary (*Mansour et al., 2008*).

In our study, multivariate analysis of preoperative variables showed that OS and DFS were significantly worse in patients with stage cT4 and subgroup analysis shows that neoadjuvant chemotherapy could not improve the prognosis of these patients. Our finding was somewhat different from the conclusion of other study that neoadjuvant treatment for patients with stage T4 have decreased rates of positive surgical margins and improved OS (*Towe et al., 2021*). The reason for this difference may be due to the inclusion of patients with stage T4N2 and no patients with stage T4N0 in our study according to the result that neoadjuvant chemotherapy could prolong the survival of patients with stage T4N0-1 (*Hamouri et al., 2023*). In brief, to further substantiate the role of neoadjuvant chemotherapy in patients with stage T4 requires validation through large-scale randomized controlled trials.

Previous investigators have shown that neoadjuvant chemotherapy can significantly improve the survival of patients (*Rosell et al., 1994*; *Josephides et al., 2024*; *Shin et al., 2024*). On the contrary, a retrospective study of 1,033 patients with stage IIIA underwent pneumonectomy showed that neoadjuvant chemotherapy did not prolong the survival of patients (*Broderick et al., 2016*). The drawback of this study was it did not include the patients with stage II, because a randomized phase III study reported that neoadjuvant chemotherapy had a significant effect on progression-free survival in those patients (*Scagliotti et al., 2012*). Our study included patients with clinical stage II-III, and the results showed that the 5-year OS and 5 year DFS were significantly increased in the neoadjuvant chemotherapy group. In further multivariate analysis, neoadjuvant chemotherapy also significantly improved the OS and DFS of patients.

However, our study has several limitations. Due to the inherent constraints of retrospective research, we were unable to ascertain the specific reasons why some patients did not undergo surgery following neoadjuvant therapy. We fully recognize that the proportion of patients who could not proceed to surgery due to treatment-related toxicities or perioperative outcomes (including mortality and complications) is crucial for evaluating the risk-benefit balance of this approach. To minimize this potential bias, we rigorously excluded all patients who experienced perioperative mortality during the screening and enrollment process. Therefore, we have reasonable grounds to conclude that neoadjuvant chemotherapy demonstrates a beneficial effect on improving prognosis in patients undergoing pneumonectomy. Nevertheless, these findings require further validation through multicenter randomized controlled trials.

While we acknowledge that performing propensity score matching (PSM) before logistic regression would enhance the evidentiary level of our study, the retrospective design imposed substantial limitations. Specifically, the current sample size was insufficient to support robust PSM analysis. We anticipate that future studies with larger cohorts employing PSM and more comprehensive analyses will provide stronger scientific validation of our findings.

## CONCLUSION

Neoadjuvant chemotherapy plays a positive role in improving the prognosis of patients with locally advanced central non-small cell lung cancer undergoing pneumonectomy, Multivariate analysis shows that neoadjuvant chemotherapy can significantly improve patient survival. Patients with downstaging after neoadjuvant chemotherapy appear to have a better survival. As has been pointed out, for patients with stage cN2, neoadjuvant chemotherapy is associated with a better survival.

### Funding
This study was funded by Hebei Province–Government Funded Clinical Excellence Program. The funders had no role in study design, data collection and analysis, decision to publish, or preparation of the manuscript.

### Grant Disclosures
The following grant information was disclosed by the authors:
Hebei Province–Government Funded Clinical Excellence Program.

### Competing Interests
The authors declare that they have no competing interests.

### Author Contributions
- Yuchen Wang conceived and designed the experiments, performed the experiments, analyzed the data, prepared figures and/or tables, authored or reviewed drafts of the article, and approved the final draft.
- Zhihang Dang performed the experiments, analyzed the data, authored or reviewed drafts of the article, and approved the final draft.
- Pu Jiang performed the experiments, prepared figures and/or tables, and approved the final draft.
- Zhifeng Li performed the experiments, prepared figures and/or tables, and approved the final draft.
- Jin Yang performed the experiments, prepared figures and/or tables, and approved the final draft.
- Kun Gao performed the experiments, analyzed the data, prepared figures and/or tables, and approved the final draft.

- Xiaona Chen performed the experiments, authored or reviewed drafts of the article, and approved the final draft.
- Jifang Yao conceived and designed the experiments, performed the experiments, authored or reviewed drafts of the article, and approved the final draft.

## Human Ethics

The following information was supplied relating to ethical approvals (*i.e.*, approving body and any reference numbers):

This study was approved by the ethics committee of The Fourth Hospital of Hebei Medical University. (2022MECD58)

## Data Availability

The raw measurements are available in the Supplemental File.

## Supplemental Information

Supplemental information for this article can be found online at http://dx.doi.org/10.7717/peerj.20007#supplemental-information.

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
