# Peer review of "A retrospective analysis of neoadjuvant chemotherapy in pneumonectomy for locally advanced central non–small cell lung cancer"

_PeerJ, doi:10.7717/peerj.20007_

## Round 0.1 · original submission · Major Revisions

**Language Note:** The review process has identified that the English language must be improved. PeerJ can provide language editing services - please contact us at [email protected] for pricing (be sure to provide your manuscript number and title). Alternatively, you should make your own arrangements to improve the language quality and provide details in your response letter. – PeerJ Staff

Reviewer 1 ·

Basic reporting

This study described the efficacy of neoadjuvant chemotherapy in patients with locally advanced NSCLC. Overall, the study is well designed. The English should be thoroughly edited, and the quality of images should be improved.

Experimental design

The study enrolled a total number of 104 patients with locally advanced NSCLC and classified into neoadjuvant group and non-neiadjuvant group.
1) In Table 1, the authors summarized clinical characteristics of the participants. It is worthy to note the patients in neoadjuvant group (n=69) was approximately 2-fold than that in the non-neoadjuvant group (n=35), and patients with T2 disease were more prevalent in the neoadjuvant group. The bias in distribution patterns may have an influence in the therapeutic outcomes.
2) In Lines 128 and 129, ciaplatin was used at the dosage of 15mg/m2. Please check if this was a typo error.

Validity of the findings

1) In Lines 158-159, pCR was not achieved in both groups. Please specift if there were patients achieved MPR?
2) In Line 178, the authors stated the 5-year DFS ratio of the stages II-III patients was 0% in both groups, which seemed to be in sharp contrast to other studies.
3) Since this is a retrospective study enrolling patients before 2019, in a time period when immunotherapy was not available in China. However, there are growing evidence showing the efficacy of immunotherapy in locally advanced NSCLC, in both neoadjuvant and adjuvant settings. Chemotherapy may not be considered as a sole strategy to downstage locally advanced disease. To this end, the significance of this study was limited.

·

Basic reporting

Reference 19,line 205. The ALITA study focuses on postoperative adjuvant therapy and has little relevance to the neoadjuvant therapy discussed in this article.

The author studied the postoperative effect of neoadjuvant chemotherapy on pneumonectomy. However, only chemotherapy has been studied, but in recent years, new adjuvant treatment schemes such as chemotherapy and immune drugs have begun to become the mainstream of lung cancer treatment, and the treatment effect is good.

line 248.'On the country' should be changed to 'on the contrary'.

Experimental design

no comment

Validity of the findings

Neoadjuvant therapy improved patients' OS and PFS, but the author did not mention the proportion of neoadjuvant patients who did not undergo surgery because of the side effects of neoadjuvant therapy, nor did he provide perioperative death and complications data. It is difficult to understand the risks of neoadjuvant therapy.

Most of the references are articles from 10 years ago, and more recent documents need to be added.

---

## Round 0.2 · accepted · Accept

The concerns raised in the previous round of review have been reasonably addressed.